5-Fluorouracil and irinotecan (SN-38) have limited impact on colon microbial functionality and composition in vitro

Vanlancker Eline 1
Vanhoecke Barbara 1
Stringer Andrea 2
Van de Wiele Tom tom.vandewiele@ugent.be 1
1 Center for Microbial Ecology and Technology (CMET), Ghent University , Ghent , Belgium
2 Sansom Institute for Health Research, University of South Australia , Adelaide , Australia
Smidt Hauke
Electronic publication date: 2017 Nov 16
Publication date: 2017
Volume: 5
Electronic Location ID: e4017
Received 2017 Jul 13; Accepted 2017 Oct 20
Copyright: ©2017 Vanlancker et al.
Copyright year: 2017
Copyright holder: Vanlancker et al.
License: This is an open access article distributed under the terms of the Creative Commons Attribution License, which permits unrestricted use, distribution, reproduction and adaptation in any medium and for any purpose provided that it is properly attributed. For attribution, the original author(s), title, publication source (PeerJ) and either DOI or URL of the article must be cited.
License URL: https://creativecommons.org/licenses/by/4.0/

Keywords: Irinotecan, Chemotherapy, Colon microbiota, 5-Fluorouracil, Community analysis, Gut, Bacteria, Short chain fatty acids, 16S rRNA sequencing, Illumina amplicon sequencing

Funding: Bijzonder Onderzoeksfonds BOF13/DOC/280 Ghent University BOF17/GOA/032 Seventh Framework Programme 299169 CAESIE: Connecting Australian-European Science and Innovation Excellence Priming Grant Eline Vanlancker is a doctoral research fellow supported by the Bijzonder Onderzoeksfonds (Ghent University; BOF13/DOC/280). This research was funded by BOF17/GOA/032 from Ghent University. Barbara Vanhoecke was funded by the Seventh Framework Programme (FP7/2011) under grant agreement no 299169 (Mucositis Platform). Andrea Stringer was supported by a CAESIE: Connecting Australian-European Science and Innovation Excellence Priming Grant. The funders had no role in study design, data collection and analysis, decision to publish, or preparation of the manuscript.

==============================
Gastrointestinal mucositis is a debilitating side effect of chemotherapy treatment, with currently no treatment available. As changes in microbial composition have been reported upon chemotherapy treatment in vivo, it is thought that gut microbiota dysbiosis contribute to the mucositis etiology. Yet it is not known whether chemotherapeutics directly cause microbial dysbiosis, thereby increasing mucositis risk, or whether the chemotherapeutic subjected host environment disturbs the microbiome thereby aggravating the disease. To address this question, we used the M-SHIME®, an in vitro mucosal simulator of the human intestinal microbial ecosystem, as an experimental setup that excludes the host factor. The direct impact of two chemotherapeutics, 5-fluorouracil (5-FU) and SN-38 (active metabolite of irinotecan), on the luminal and mucosal gut microbiota from several human donors was investigated through monitoring fermentation activity and next generation sequencing. At a dose of 10 µM in the mucosal environment, 5-FU impacted the functionality and composition of the colon microbiota to a minor extent. Similarly, a daily dose of 10 µM SN-38 in the luminal environment did not cause significant changes in the functionality or microbiome composition. As our mucosal model does not include a host-compartment, our findings strongly indicate that a putative microbial contribution to mucositis is initially triggered by an altered host environment upon chemotherapy.

Introduction

5-Fluorouracil (5-FU) and irinotecan (active metabolite: SN-38) are two commonly used chemotherapeutic agents used for cancer treatment. A major side effect of these agents is gastrointestinal mucositis, an inflammation and ulceration of the gastrointestinal mucosa, mostly diagnosed based on the occurrence of diarrhea (Benson et al., 2004; Peterson et al., 2015). The incidence of chemotherapy-induced diarrhea associated with 5-FU or irinotecan treatment varies around 50–80% (Benson et al., 2004). Treatment of colon cancer with FOLFOX (5-FU, leucovorin, and oxaliplatin) or FOLFIRI (5-FU, leucovorin and irinotecan) or IROX (Irinotecan and oxaliplatin) results in a risk for grade 3 or 4 diarrhea (severe diarrhea requiring hospitalization or having life-threatening consequences, according to the National Cancer Institute Common Toxicity Criteria of Adverse Events (NCI CTCAE) classification) of 10%, 10% and 24%, respectively (Keefe et al., 2007). While gastrointestinal mucositis often causes cessation of the cancer treatment and a lot of discomfort for the patient, no effective treatment is available yet. The Multinational Association of Supportive Care in Cancer (MASCC) has put an effort into formulating guidelines and recommendations on how to prevent and treat gastrointestinal mucositis, such as by the use of octreotide for the treatment of diarrhea associated with hematopoietic stem cells transplantation (Lalla et al., 2014).

In research on the pathobiology of mucositis more and more interest is going to the role and/or the effect of the gut microbiota during chemotherapy treatment (Stringer, 2013; Van Vliet et al., 2010). Microbiota have an important function in many pathways some of which are also involved in the development of gastrointestinal mucositis. For example, microbiota influence intestinal permeability and thickness of the mucus layer, both important in barrier function during mucositis (Touchefeu et al., 2014; Van Vliet et al., 2010). Next, recognition of microbial antigens to toll-like receptors (TLR) can activate the nuclear factor-kappa B (NF-κB) pathway, resulting in production of pro-inflammatory cytokines that are crucial in the mucositis pathobiology (Touchefeu et al., 2014; Van Vliet et al., 2010; Vanhoecke & Stringer, 2015). Under healthy conditions, commensal microorganisms are thought to live in a homeostatic relationship with the host providing a low grade immune activation and stimulating epithelial repair in an NF- κB dependent pathway (Touchefeu et al., 2014; Van Vliet et al., 2010). But during cancer treatment, changes at the level of the host and/or microbiome may disturb this homeostatic relationship and increase the inflammatory status.

Animal and human studies have repeatedly shown that chemotherapeutics can change the gut microbiota (Stringer, 2013; Touchefeu et al., 2014). In rat models, both 5-FU (Stringer et al., 2009a; Von Bultzingslowen et al., 2003) and irinotecan (Lin et al., 2012; Stringer et al., 2009b; Stringer et al., 2007; Stringer et al., 2008) modified the gut microbiome, with a decrease in commensal microbiota and increases in Escherichia spp., Clostridium spp. and Enterococcus spp. Clinical studies report on shifts of faecal microbiota upon chemotherapy treatment (Touchefeu et al., 2014), with a lower diversity and total number of microbiota observed after chemotherapy treatment. Most frequent changes are decreases in Bifidobacterium, Faecalibacterium and Clostridium cluster XIVa and increases in Bacteroides and Escherichia (Montassier et al., 2014; Stringer et al., 2013b; Van Vliet et al., 2009; Zwielehner et al., 2011). Yet besides impacting the microbiota, chemotherapeutics also affect the mucus layer and the number of goblet cells which is likely the result of an increase in pro-inflammatory cytokine levels and leading to an altered mucosal barrier (Stringer, 2013).

To further unravel the role of microbiota in gastrointestinal mucositis, it is important to understand the direct impact of chemotherapeutics on the functionality and composition of the gut microbiome. While animal and human studies do not allow to distinguish between host and microbiome effects, the goal of the present study was to investigate the direct effect of two chemotherapeutics, 5-FU and SN-38, on the gut microbiome. Therefore, we used an in vitro model that was proven to be representative for the human colon microbiome: the M-SHIME® model (Mucosal-Simulator of the Human Intestinal Microbial Ecosystem) (Van den Abbeele et al., 2012).

Materials and Methods

Chemicals

A filter-sterilized stock solution of 10 mM 5-Fluorouracil (5-FU) (Sigma Aldrich, Diegem, Belgium) was prepared in dimethyl sulfoxide (DMSO). The stock solution was further diluted (1:1,000) in the mucin agar (see M-SHIME Experimental set-up) to a final concentration of 10 µM.

A filter-sterilized stock solution of 10 mM SN-38 (Sigma Aldrich, Diegem, Belgium) was prepared in DMSO. The stock solution was further diluted (1:1,000) in the medium to a final concentration of 10 µM.

M-SHIME

Experimental set-up

The M-SHIME® (Mucosal-Simulator of the Human Intestinal Microbial Ecosystem, joint registered name from Ghent University and ProDigest) is an in vitro dynamic model for the human intestinal tract, incorporating both luminal and mucosal colon environment, resulting in distinct luminal and mucosal microbial populations (Van den Abbeele et al., 2012). The set-up used in this study consisted of a stomach/small intestine vessel and two proximal colon vessels (control and treatment) for six human donors in parallel (Fig. 1).

Figure 1 Experimental set-up.

(A) M-SHIME with 5-FU and (B) M-SHIME with SN-38. Arrows indicate the time of dosing.

Faecal samples were collected and prepared within 1 h according to standard procedures (Molly, Woestyne & Verstraete, 1993). In short, aliquots (20 g) of fresh faecal samples were diluted and homogenized with 100 mL 0.1 M phosphate buffer (8.8 g K2HPO4 L−1 and 6.8 g KH2PO4 L−1, pH 6.8) containing 1 g L−1 sodium thioglycolate as reducing agent. After removal of the particulate material by centrifugation (2 min, 500 g), each colon vessel was inoculated with 40 mL of the faecal suspension.

The double-jacketed vessels were kept at 37 °C and flushed daily with N2 (5 min) to assure anaerobic conditions. All colon compartments (500 mL) were stirred (200 rpm) and pH-controlled (pH 5.6–5.9). Mucosal conditions were created as described by Van den Abbeele et al. (2012). Briefly, 80 mucin agar-covered microcosms (AnoxKaldnes K1 carrier; AnoxKaldnes AB, Lund, Sweden) in a polyethylene netting were added to each vessel (Zakkencentrale, Rotterdam, the Netherlands) (Van den Abbeele et al., 2012). Half of the microcosms were replaced every other day with fresh sterile ones under a flow of N2 to maintain anaerobic conditions. After washing the beads twice with Phosphate Buffered Solution to discard luminal microbiota, samples of the mucin-agar were taken and stored at −20 °C.

After an initial incubation of 16 h, pumps were switched on in order to supply each colon compartment with 140 mL nutritional medium and 60 mL pancreatic juice three times a day. The nutritional medium contained (in g/l) yeast extract (3.0) (Oxoid), special peptone (1.0) (Oxoid), mucin (2.0) (Sigma), arabinogalactan (0.25) (Sigma), pectin fromapple (0.5) (Sigma), xylan (0.25) (Sigma), potato starch (1.0) (Anco). The pancreatic juice was prepared as described earlier by Van den Abbeele et al. (2012). The treatment started after two days of stabilisation.

Experimental design M-SHIME with 5-FU

In this set-up we compared a challenge of the in vitro cultured human microbiota with 5-FU to a control situation (Fig. 1A). As 5-FU is intravenously supplemented to the patient and reaches the gut through the mucus layer, we chose to dose 5-FU directly in the mucus-covered microcosms. Taking into account pharmacokinetics during continuous infusion with 5-FU, a representative serum concentration range of 5-FU in vivo is 3–10 µM. We chose the highest concentration to evaluate the effect to mucosal 5-FU towards the gut microbiome. DMSO was used as a negative control. On day 0, 2 and 4, half of the mucin-covered microcosms were replaced with new treated ones. Therefore, three doses of 5-FU were given in total. To cover biological reproducibility, we compared six different donors, all healthy volunteers who had no history of antibiotic treatment up to six months prior to the study (Ethical approval from Ghent University hospital, Belgian Registration number BE 6700201214538; verbal consent). Luminal samples were taken every 24 h and mucosal samples every 48 h (on day 2, 4 and 6). For donor 1 and 2, additional luminal samples were taken 6 h after each treatment. All samples were stored at −20 °C.

Additional distal colon vessels were run for donor 1 and 2 (control and treatment). The distal colon vessels (800 ml) were connected to the respective proximal colon vessel, treated similarly with 5-FU (or DMSO as a control) via the mucin beads and pH-controlled at pH 6.6–6.9.

Experimental design M-SHIME with SN-38

In this set-up we compared a challenge of the in vitro cultured human microbiota with SN-38 to a control situation (Fig. 1B). As SN-38 enters the small intestine via the bile, the treatment was added daily to the luminal environment, just before the feed entered the colon compartments. An average colon concentration of 1–2 µM was estimated based on faecal concentrations of SN-38. Therefore, a concentration of 10 µM was used to make sure to test the highest physiological relevant concentration. The M-SHIME system was treated with SN-38 for 6 consecutive days. DMSO without SN-38 was used as a negative control. On day 0, 2 and 4, half of the mucin-covered microcosms were replaced with new ones (non-treated). To cover biological reproducibility, we compared five different donors (healthy volunteers) who had no history of antibiotic treatment up to 6 months prior to the study (Ethical approval from Ghent University hospital, Belgian Registration number BE 6700201214538). Both luminal and mucosal samples were taken every 48 h. All samples were stored at −20 °C. Additional distal colon vessels were run for donor 1 (control and treatment). The distal colon vessels (800 ml) were connected to the respective proximal colon vessel and pH-controlled at pH 6.6–6.9.

Moreover, a luminal-SHIME (L-SHIME) was run for donor 1 (in duplicate). This set-up is identical to the normal set-up with the proximal colon vessels, but does not include a mucus compartment. Therefore, 4.0 g/l mucin was added to the nutritional medium instead of 2.0 g/l.

Short chain fatty acids (SCFA)

Luminal samples were diluted 1:2 to a total volume of 2 ml and the SCFA were extracted with diethyl ether and analysed using a gas chromatograph as described by De Weirdt et al. (2010). 2-methyl hexanoic acid was used as an internal standard. The concentration of acetate, propionate, butyrate, isobutyrate, valerate, isovalerate, caproate and isocaproate was determined in each sample and the total amount of SCFA was calculated as the sum of all. The relative concentration of each SCFA was expressed as mol% being the ratio of its concentration (mM) and the total SCFA concentration (mM) multiplied by 100.

DNA extraction

Luminal samples (1 mL) for total DNA extraction were centrifuged for 10 min at 18,000 rcf, supernatant was removed and the pellet was stored immediately at −20 °C until further analysis.

Total DNA was extracted from the pellet of 1 mL liquid samples or 0.25 g mucin-agar according to a protocol adapted from Vilchez-Vargas et al. (2013). Cells were lysed with 1 mL lysis buffer (100 mM Tris/HCl pH 8.0, 100 mM EDTA pH 8, 100 mM NaCl, 1% (m/v) polyvinylpyrrolidone and 2% (m/v) sodium dodecyl sulphate) and 200 mg glass beads (0.11 mm, Sartorius) in a FastPrep® 96 instrument (MP Biomedicals, Santa Ana, CA, USA) for two times 40 s (1,600 rpm). Following removal of glass beads by centrifugation (5 min at 18,000 rcf), DNA was extracted from supernatant using a phenol–chloroform extraction (Vilchez-Vargas et al., 2013). The DNA was precipitated at −20 °C with 1 volume of ice-cold isopropyl alcohol and 0.1 volume of 3 M sodium acetate for at least 1 h. After removal of isopropyl alcohol by centrifugation (30 min, 18,000 rcf), the DNA pellet was dried and resuspended in 100 µL 1x TE (10 mM Tris, 1 mM EDTA) buffer. The DNA samples were immediately stored at −20 °C until further analysis. The quality of DNA samples was analysed by gel electrophoresis (1.2% (w/v) agarose) (Life technologies, Madrid, Spain). The DNA samples were diluted (1:10) for further analysis.

Microbial community analysis

Denaturing Gradient Gel Electrophoresis (DGGE) was performed as described in Supplementary Information (S14).

16S rRNA gene amplicon sequencing was performed by LGC Genomics (Berlin, Germany) on the Illumina MiSeq platform. The V3–V4 region of the 16S rRNA gene was amplified using primers derived from Klindworth et al. (2013): 341F (NNNNNNNNNTCCTACGGGNGGCWGCAG) and 785R (NNNNNNNNNNTGACTACHVGGGTATCTAAKCC). The PCR mix included 1 µl of DNA extract, 15 pmol of both the forward and reverse primer in 20 µL of MyTaq buffer containing 1.5 units MyTaq DNA polymerase (Bioline) and 2 µl of BioStabII PCR Enhancer (Sigma). For each sample, the forward and reverse primers had the same unique 10-nt barcode sequence. The PCR program consisted of 2 min 96 °C predenaturation and 20 cycli of 96 °C for 15 s, 50 °C for 30 s, 70 °C for 90 s. Next, ∼20 ng amplicon DNA (determined by gel electrophoresis) of each sample was pooled for up to 48 samples carrying different barcodes. The amplicon pools were purified with one volume AMPure XP beads (Agencourt, Beverly, MA, USA) to remove primer dimer and other small mispriming products, followed by an additional purification on MinElute columns (Qiagen, Hilden, Germany). Finally, about 100 ng of each purified amplicon pool DNA was used to construct Illumina libraries by means of adaptor ligation using the Ovation Rapid DR Multiplex System 1–96 (NuGEN, San Carlos, CA, USA). Illumina libraries were pooled and size selected by preparative gel electrophoresis. Sequencing was done on an Illumina MiSeq using v3 Chemistry (Illumina, Hayward, CA, USA). To assess the sequencing quality a mock community was included in triplicate in the sequencing run (error rate = 0.183%).

The Mothur software package (v.1.33.3), and guidelines developed by Schloss, Gevers & Westcott (2011) were used to process Illumina data. Forward and reverse reads were assembled into contigs and ambiguous contigs or contigs with divergent lengths were removed. The number of unique sequences was determined and these were aligned to the Mothur-reconstructed SILVA Seed alignment (v123). Sequences not aligning within the region targeted by the primer set or sequences with homopolymer stretches with a length higher than 12 were removed. Sequences were pre-clustered together within a distance of 1 nucleotide per 100 nucleotides. These cleaned-up and preclustered sequences were checked for Chimera’s (using Uchime). The sequences were classified using RDP (Ribosomal Database Project) release 14 and a naive Bayesian classifier (Wang’s algorithm). All sequences that were classified as Eukaryota, Archaea, Chloroplasts and Mitochondria were removed. If sequences could not be classified at all (even not at (super)Kingdom level) they were removed. OTU’s were clustered with an average linkage and at the 97% sequence identity. The sequences reported in this paper have been deposited in the European Nucleotide Archive (ENA) database (Accession number LT800946–LT802885).

Statistical analysis

All statistical analyses were performed in R (version 3.3.2) (R Development Core Team, 2016).

Statistical inference on the 5-FU and SN-38 treatment effect on the SCFA concentration was performed by spline regression. Natural splines were fitted for each group (control and treatment) to the scaled and centered temporal data because these provide more stable estimates at the boundary time points (James et al., 2014). Knots were fixed at the 33.3% and 66.6% quantiles. Model parameter estimation was performed by the ordinary least squares method, resulting in model residuals that were normal distributed and did not exhibit temporal autocorrelation. Due to the presence of moderate heteroscedasticity in the model residuals, robust White heteroscedasticity-consistent standard errors (vcovHC function, sandwich v2.3-4 package) were used in the statistical inference on the treatment effect (type II ANOVA).

The packages phyloseq (McMurdie & Holmes, 2013) and vegan (Oksanen et al., 2016) were used for microbial community analysis. Heatmaps were generated with the pheatmap package and order-based Hill’s numbers (Hill, 1973) were calculated. If the data were normally distributed (tested with Shapiro–Wilk test) and homoscedastic (tested with Levene test), differences in Hill’s numbers were defined via ANOVA and Tukey as post-hoc test; if not, Kruskall Wallis test with Tukey post-hoc testing was used. Non-metric distance scaling (NMDS) plots of the bacterial community data were created based on the Bray-Curtis distance measures. Significant differences were identified by means of Permutational ANOVA (PERMANOVA) using the adonis function (vegan).

Results

Prior to the evaluation of the effect of 5-FU and irinotecan, we evaluated the overall metabolic activity and community composition of our SHIME runs. For both runs, the microbial fermentation activity of the proximal colon (short-chain fatty acid production) and community composition (ratio Firmicutes/Bacteroidetes/Proteobacteria, Tables S1 and S2) was shown to be consistent with that of previous SHIME runs (Geirnaert et al., 2015; Molly et al., 1994). This convinced us that the SHIME runs were executed in a proper and standardised way.

Effect of 5-FU on the metabolic activity in the gut

The effect of 5-FU (at 10 µM in the mucosal environment) on the gut microbiome of 6 healthy donors was investigated using a M-SHIME® harbouring both luminal and mucosal microbiota (Fig. 1A). SCFA analysis showed interindividual differences and the total luminal SCFA concentration before the first treatment ranged from 25.5 µM to 39.5 µM between the six donors. Overall, there is no significant difference between treatment with 5-FU and control behaviour through time (p = 0.18). Although, for donor 1 and donor 2, total SCFA concentrations started increasing after the second 5-FU treatment, compared with the control, with a final increase of 113% and 76% respectively at day 6 (Fig. 2). The same trends (i.e., an increase for donor 1 and 2 and no effect for all other donors), were observed for the total mucosal SCFA concentration (Fig. S1). Further, 5-FU did not have any effect on the relative proportions of acetate, propionate, butyrate or branched SCFA (Fig. S2). For donor 1 and 2, the experimental set-up also allowed the inclusion of a distal colon region (pH 6.6–6.9). However, we observed no effect of 5-FU on the SCFA production (Fig. S3).

Figure 2 Microbial functionality of M-SHIME with 5-FU for donors 1, 2, 3, 4, 5 and 6 (figures A, B, C, D, E and F, respectively).

At 10 mM in the mucosal environment of the M-SHIME, 5-fluorouracil increases luminal total short chain fatty acid concentrations for donor 1 and 2 but has no overall effect for all donors (p = 0:18). Arrows indicate the time of dosing.

Effect of 5-FU on the gut microbial profile

To investigate the effect of 5-FU on the composition of both luminal and mucosal microbiota, both denaturing gradient gel electrophoresis (DGGE) and amplicon sequencing of the 16S rRNA gene were performed for all samples at day 0 and day 6.

Clustering analysis of DGGE profiles showed prominent interindividual variability, but no significant impact of 5-FU treatment. Even after 6 days of treatment, profiles still clustered together for each donor with similarities between control and treatment ranging from 79–98% for luminal samples and 93% to 99% for mucosal samples, except for donor 2 who had only 70% similarities. Hence, there was no clear shift in the microbial profile for all donors, only minor differences between control and 5-FU treated samples could be observed (Fig. S4).

16S rRNA gene amplicon sequencing showed that all samples were dominated by Proteobacteria, Bacteroidetes and Firmicutes. At the genus level, Escherichia/Shigella, Bacteroides, Veillonella and Clostridium cluster XIVa were most abundant (Figs. 3 and S5). Also, NMDS plots showed no significant difference between control and 5-FU treatment (p = 0.75) and treatment condition only explained 2.7% of the variation of the microbiome after treatment (based on Bray–Curtis dissimilarities on OTU level) (Fig. 4A). For all samples, donor individuals and sampling timepoint explained most of the variation, respectively 38.9% and 14.8% (p = 0.0001 and p = 0.0001) (Table 1 and Fig. S6).

Figure 3 Microbial community analysis of M-SHIME with 5-FU in the simulated gut lumen (A) and simulated gut mucus (B) environment.

Heatmap representing the most abundant genera (at least 0.1% on average) showed no clear effect of treatment with 5-FU on the gut microbial composition as determined by 16S rRNA gene amplicon sequencing.

Figure 4 Microbial community composition of M-SHIME at day 6.

NMDS plots based on Bray Curtis dissimilarities of 16S rRNA gene amplicon sequencing data at day 6 (after treatment) showed large interindividual variability (5-FU: p = 0.0001, R2 = 70.1%; SN-38: p = 0.0001, R2 = 44.3%), but no clear effect of the treatment with (A) 5-FU (p = 0.75, R2 = 2.7%) or (B) SN-38 (p = 0.31, R2 = 5.1%).

At the genus level, the 5-FU treatment showed a trend in increased relative abundance of Bacteroides from 24.4 ± 13.2% to 41.1 ± 11.6% (p = 0.065) and decreased abundance of Escherichia/Shigella from 19.7 ± 17.1% to 8.9 ± 7.2% (p = 0.23) in the lumen for 5 out of 6 donors. In the mucus fraction, these shifts could not be observed. In contrast, 5-FU did not influence the diversity in the lumen at the end of treatment, but did increase the diversity (first and second order Hill numbers) in the mucus (p = 0.010 and p = 0.055 respectively), indicating a higher evenness and diversity (Fig. S7). In general, mucus samples had a higher diversity than luminal samples (p-values for Hill numbers: p = 0.058, p = 0.0032 and p = 0.0039) (Fig. S7). Only for donor 2, a changed mucus bacterial profile was observed, with increases in Anaeroglobus, Roseburia and Parabacteroides (Figs. 3 and S5).

Effect of SN-38 on the metabolic activity in the gut

The effect of SN-38 (the active metabolite of irinotecan) (at 10 µM in the luminal environment) on the gut microbiome of five healthy donors was investigated using a M-SHIME with proximal colon vessels (Fig. 1B). Interindividual differences in the total luminal SCFA concentration before the first treatment ranged from 18.3 µM to 45.9 µM between the five donors. Overall, there is no significant difference between treatment with SN-38 and control behaviour through time (p = 0.85) (Fig. 5). A similar trend (no effect of SN-38) was observed for the total mucosal SCFA concentrations (Fig. S8). Also, for the relative luminal concentrations of acetate, propionate, butyrate and branched SCFA, no impact of SN-38 was detected (Fig. S9). For donor 1 (in duplicate), the experimental set-up allowed the inclusion of a distal colon region (pH 6.6–6.9). However, we observed no effect of SN-38 had on the SCFA levels of the distal colon microbiota (Fig. S10). Similarly, the experimental set-up also allowed running a luminal-SHIME system (without mucus compartment). However, again no effect on SCFA production was observed, excluding the fact that the mucuslayer protects the microbiota against the SN-38 (Fig. S11).

Table 1 Confouding factors explaining variation in microbial community composition.

P-values and R for different confounding factors based on Bray-Curtis dissimilarities of the 16S rRNA gene Illumina amplicon sequencing data (significant values are indicated in italic).

		Donor	Time	Lumen/mucus	Treatment	
5-FU					
All	p-value	0.0001	0.0001	0.0071	0.90	
	R2	38.9%	14.8%	5.2%	1.2%	
Day 0	p-value	0.0001	NA	0.033	0.96	
	R2	63.8%	NA	8.7%	1.8%	
Day 6	p-value	0.0001	NA	0.16	0.75	
	R2	70.1%	NA	6.5%	2.7%	
SN-38					
All	p-value	0.0097	0.0001	0.0014	0.72	
	R2	17.7%	30.3%	9.0%	1.4%	
Day 0	p-value	0.0001	NA	0.0028	0.77	
	R2	54.2%	NA	13.7%	2.7%	
Day 6	p-value	0.0001	NA	0.0001	0.31	
	R2	44.3%	NA	24.0%	5.1%	

Figure 5 Microbial functionality of M-SHIME with SN-38 for donors 1, 2, 3, 4, 5 and 6 (A, B, C, D, E and F, respectively).

At 10 mM in the luminal part of the M-SHIME, irinotecan (SN-38) has no effect on luminal total short chain fatty acid concentrations (p = 0:85). Arrows indicate the time of dosing.

Effect of SN-38 on the gut microbial profile

To investigate the effect of SN-38 on the composition of both luminal and mucosal microbiota, DGGE and amplicon sequencing of the 16S rRNA gene were performed for all samples at day 0 and day 6.

Clustering analysis of DGGE profiles showed large interindividual variability, but no effect of SN-38 treatment was observed. After six days of treatment microbial profiles still clustered together for each donor with similarities ranging from 94 to 98% for luminal samples (even higher than at the start) and 68 to 92% for mucosal samples. There were no clear shifts in the microbial profile of all five donors and only minor differences between control and SN-38 treated samples could be observed (Fig. S12).

16S rRNA gene amplicon sequencing showed that all samples were dominated by Proteobacteria, Bacteroidetes and Firmicutes. At the genus level, Escherichia/Shigella, Bacteroides, Veillonella and Clostridium cluster XIVa were most abundant (Fig. 6 and S13). The NMDS plots demonstrated no significant difference between control and SN-38 treatment (p = 0.31) and the treatment condition only explained 5.1% of the variation of the microbiome after treatment (based on Bray–Curtis dissimilarities on OTU level) (Fig. 4B). For all samples, donor individuals and sampling timepoint explained most of the variation, respectively 17.7% and 30.3% (p = 0.0097 and p = 0.0001) (Table 1 and Fig. S6).

Figure 6 Microbial community analysis of M-SHIME with SN-38 in the simulated gut lumen (A) and simulated gut mucus (B) environment.

Heatmap representing the most abundant genera (at least 0.1% on average) showed no clear effect of treatment with SN-38 on the gut microbial composition as determined by 16S rRNA gene amplicon sequencing.

No effect of SN-38 on specific genera or diversity indices were noticed for all donors, but some donor-specific changes of relative abundances of some genera could be observed. For donor 2 and 3, an increase in Cloacibacillus was observed (from respectively 1.4 to 14.6% and 0 to 22.4%) in presence of SN-38 in the lumen, compared with the control. For donor 3 and donor 5, Alistipes increased in the lumen in presence of SN-38 compared with the control (from 0.4 to 9.4% and 0.09 to 7.8% respectively). In the mucus, an increase in Roseburia for donor 1 (in duplicate) and donor 2 was observed in presence of SN-38 compared with the control (from 9.9 to 24.9%, 2.4 to 25.1% and 2.1 to 18.5% respectively) (Figs. 6 and S13). Similar to 5-FU, mucosal samples showed a higher evenness compared with luminal samples (Hill number order 1, p = 0.0094) (Fig. S7).

Discussion

Patients often have to deal with several side effects from cancer treatment, of which gastrointestinal mucositis is one of the most debilitating. The major symptoms are inflammation and ulceration of the gastrointestinal tract accompanied with diarrhoea. Gut microbiota are more and more believed to play a role in the etiology and severity of mucositis, but the direct effect of chemotherapy on gut microbiota is not yet clear. In this study, we show that 5-FU and SN-38 only have a limited impact on colon microbial functionality and composition in an in vitro model mimicking both the luminal and mucosal gut microbiota.

5-FU was added solely to the mucosal part of the M-SHIME® as it reaches the gut mainly via the blood and gut mucosa. Plasma concentrations of 5-FU can reach some hundreds of µM in cancer patients, for example with a bolus injection with 5-FU, but they rapidly drop to 15–30 µM after 30 min and to 0 µM after 2 h (Casale et al., 2004; Kosovec et al., 2008), due to the short half life time (6–22 min) of 5-FU (Bocci et al., 2000). In contrast, for a continuous infusion with 5-FU, the plasma concentrations will be much lower, ranging from 3 to 10 µM, but are steady for a longer time period (24 h) (Joulia et al., 1999; Takimoto et al., 1999). As we wanted to investigate a long term dosing, we chose to dose at 10 µM 5-FU in the mucus layer, which is in close contact with the blood circulation.

The active metabolite of irinotecan, SN-38, was added to the luminal part of an M-SHIME® as it reaches the colon via the small intestine and through microbial synthesis, hydrolyzing biliary secreted, non-toxic SN-38G to the active compound SN-38 by β-glucuronidase activity (Takasuna et al., 1996). This also explains why feces levels of SN-38 are much higher than plasma levels. Considering that on average 1.2% of the dose is excreted as SN-38 in the feces within 24 h (Slatter et al., 2000; Sparreboom et al., 1998), (an estimated average colon concentration of ∼1–2 µM), and assuming a homogeneous distribution along the colon, a dose of 10 µM SN-38 was used in the luminal part of the M-SHIME.

Neither 5-FU nor SN-38 had a significant impact on the functionality or the composition of the M-SHIME community. Only some donor-specific changes could be observed: increases in Bacteroides, Cloacibacillus, Alistipes and Roseburia and a decrease in Escherichia/Shigella after the simulated chemotherapy treatment. Our observations are in contrast with what was found in human trials analyzing stool samples after chemotherapy treatment, namely an increase in Escherichia and a decrease in Roseburia and varying trends for Bacteroides (Montassier et al., 2014; Montassier et al., 2015; Stringer et al., 2013b; Van Vliet et al., 2009; Zwielehner et al., 2011). However, these studies used different chemotherapeutic agents and the use of antibiotics was not excluded. With regards to animal studies, an increase in Escherichia was shown in fecal samples of rats treated with 5-FU (Stringer et al., 2009a).

No shifts in the M-SHIME microbiome could be observed after 5-FU treatment, although our previous research on single species clearly showed a differential sensitivity effect amongst oral species towards the drug (Vanlancker et al., 2016) and the same trend was observed for gastrointestinal microbiota (Florez et al., 2016; Stringer et al., 2009a). For example, Escherichia coli and Pseudomonas aeruginosa were highly resistant to 5-FU whereas Bifidobacteria were much more sensitive (Florez et al., 2016; Stringer et al., 2009a; Vanlancker et al., 2016). Apparently, when present in a well-balanced ecosystem as the M-SHIME system, the impact of 5-FU is very low. There is no literature information on the 5-FU metabolism by gut microbiota. For irinotecan on the other hand, 34 gut species are shown to be resistant till 330 µM irinotecan (Florez et al., 2016), although in vivo transformation of irinotecan to more toxic compounds as SN-38 had not been taken into account in this study. For both 5-FU and SN-38, however, we could not evaluate what the effect of chemotherapeutics would be on an unbalanced ecosystem, as in our study, stool samples from healthy donors (non-antibiotic exposed) were used as inoculum. Therefore, it may be interesting to investigate the effect of chemotherapeutic agents on an unbalanced gut microbial ecosystem, such as after antibiotic treatment. As many patients will also receive antibiotics during their treatment, chemotherapeutics may therefore have a bigger impact on this unbalanced, less diverse microbiome.

Although we did not see a major direct impact of chemotherapeutic treatment (5-FU and SN-38) on the colon microbiota in vitro, both clinical and animal studies show that these chemotherapy treatments can have an impact on the gut microbiota. Rat studies with 5-FU and SN-38 have shown that there are clear shifts in the microbial composition after chemotherapy treatment (Stringer et al., 2007; Touchefeu et al., 2014). We hypothesize, that host-microbe interactions or the presence of the host are needed to induce these changes in vivo and that there is only a minor direct effect of chemotherapy on the gut microbial ecosystem as such. This also suggests that the host is a major contributing element in the drug toxicity towards the gut microbiome. The stressed host environment can induce or aggravate microbial dysbiosis and indirectly increase gastrointestinal mucositis severity. Targeting the microbiome with probiotics could still be a good strategy to treat mucositis by supporting a homeostatic gut microbial ecosystem. Gut microbiota can protect the intestinal mucosa by regulating important pathways in mucositis such as TLR-NF-κB pathway, mucus layer, intestinal permeability, mucosal repair etc (Touchefeu et al., 2014).

In conclusion, 5-FU and SN-38 displayed a limited impact on microbial composition and functionality of a healthy in vitro M-SHIME® colon ecosystem. We know from clinical and animal studies that the microbiome changes upon treatment with chemotoxic agents and therefore we assume that these changes are primarily induced in presence of host cells. These modulation of the host cells and tissue can have a major impact on the gut microbiome resulting in dysbiosis that can further aggravate the mucositis process. This mechanism, where disrupted host-microbe interactions under chemotherapeutic stress contribute to the process of mucositis, needs to be further investigated.

Supplemental Information

Supplemental Information 1 Supplementary information

Click here for additional data file.

Supplemental Information 2 Raw data SCFA

Click here for additional data file.

Supplemental Information 3 Raw data—OTU table amplicon sequencing

Click here for additional data file.

Supplemental Information 4 Sequences

Click here for additional data file.

The authors would like to thank Jana De Bodt and Janie Bourgeois for the technical support, Jo De Vrieze for the support in analysing the Illumina data, Ruben Props for the statistical support and Rosemarie De Weirdt and Jo De Vrieze for their review of the manuscript.

Additional Information and Declarations

Competing Interests

Author Contributions

Human Ethics

DNA Deposition

Data Availability

The authors declare there are no competing interests.

Eline Vanlancker conceived and designed the experiments, performed the experiments, analyzed the data, wrote the paper, prepared figures and/or tables.

Barbara Vanhoecke and Andrea Stringer conceived and designed the experiments, performed the experiments, analyzed the data, reviewed drafts of the paper.

Tom Van de Wiele conceived and designed the experiments, contributed reagents/materials/analysis tools, reviewed drafts of the paper.

The following information was supplied relating to ethical approvals (i.e., approving body and any reference numbers):

The Ghent University hospital granted Ethical approval to carry out this study (Belgian Registration number BE 6700201214538).

The following information was supplied regarding the deposition of DNA sequences:

The sequences reported in this paper have been deposited in the European Nucleotide Archive (ENA) database (Accession number LT800946–LT802885).

The following information was supplied regarding data availability:

The raw data has been submitted as Supplemental Files.

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
