# Peer review of "-Fluorouracil and irinotecan (SN-38) have limited impact on colon microbial functionality and composition in vitro"

_PeerJ, doi:10.7717/peerj.4017_

## Round 0.1 · original submission · Major Revisions

Both reviewers found the study very interesting and of added value for readers of PeerJ. They also provided a number of detailed suggestions as to further improve and clarify the manuscript regarding experimental details, data analysis and presentation.

·

Basic reporting

Although the use of language is generally quite good, I would recommend asking a native English speaker to review the manuscript as some sentences are a little difficult to fully comprehend. Examples of this can be found on lines 114, and 150-152. Also, if the donors described on line 155 were the same used for the 5-FU experiment, it would be clearer if the text read ‘five of the six healthy human donors.’ and delete the remainder of the sentence.

Care should be taken with the use of abbreviations: abbreviated names are shown on lines 28-29 without previously being written, the abbreviation for irinotecan is written on line 35 but not always used later (e.g. lines 39, 41, 42), on line 58 it should read 'nuclear factor-kappa B (NF-κB)'.

Lastly, could the authors clarify whether changes described on lines 73-74 are proportional or actual numbers, consider changing line 363 to ‘after the simulated chemotherapy treatment’, and I couldn't find any reference to Figure S7 in the text.

I would like to commend the authors on Figure 1, which gives a very clear view on the experimental set-up.

Experimental design

In order to aid experimental replication, could the authors define 'mucus solution' (line 92), add suppliers/catalogue numbers to media components (lines 121-122), detail how the SCFAs were validated e.g. use of standards, or reference library (line 161), quantify the 'maximum speed' (lines 176, 180), and briefly describe or reference the phenol-chloroform method (line 177).

The research presented here is the logical next step to build on the previous research in the literature, and I strongly agree with the authors potential follow-up work described on lines 380-386.

Validity of the findings

No comment

Reviewer 2 ·

Basic reporting

The manuscript is written in an easy-to-follow English, with only a few typos and unclear statements (see section 4, general comments for specific things to change/consider). Relevant literature cited. Tables and figures generally of acceptable quality, though some improvement is needed (see section 3 for specific comments).

Experimental design

The manuscript reports primary research within the scope of the journal. Materials and methods are generally sufficiently described, though with some shortcomings (see below).

Specific comments to Materials and Methods section:
Line 92. The mucus solution is not defined (composition?)
Line 118 and elsewhere: Samples seem to be stored at -20C before further analysis. Storage beyond a few days at -20C is known to influence GM profiles as determined by high throughput amplicon sequencing. How long was the samples stored at -20C? Were some samples stored longer than others at -20C and do the authors have any data to support, that storage at this temperature did not influence the results?
Line 125-128: This information does not belong in Materials and Methods section (but rather in the results section)
Line 164-165: Lactic acid not detected? Sometimes present in rather high amounts. Please comment.
Line 176: FastPrep running at 1600 rpm? Does not make sense. Perhaps "oscilations pr. minute" is what is meant?
Line 176 and elsewhere: What is maximum speed?
Line 177: Provide more details or reference for phenol-chloroform extraction.
Line 189-206: Generally adequate details are given, but it is not clear how e.g. how the amount of 20 ng "amplicon" was determined (line 197).
Line 257-258 and 300-304 not described in Math&Meth. Leave information out?

Validity of the findings

Data generally robust, though there is some room for improvement (see below). One big concern is that it does not seem to be investigated how stable 5-FU and SN-38 is in the fermentors. One could speculate, that the main finding of the study (no direct effect of 5-FU and SN-38 on GM) could simply be a consequence of 5-FU and SN-38 being rapidly broken down. Please provide evidence (data or references) that this is not the case.

Other, smaller comments:
In the text examples of specific taxa differing significantly with treatment within the same donor are stated, but p-values are not stated in e.g. fig. 3 and 6. Further, was FDR or Bonferroni-correction carried out to avoid false positives?
Figure legends are not always as informative as they could be. P-values (and where relevant R-values) are rarely stated. It is not stated how GM-characterisation was carried out in the relevant figure legends - include information such as "as determined by 16S rRNA gene amplicon sequencing" etc.
Figure 4 is very messy and difficult to follow. Try to make it easier to understand. Perhaps by further dividing the figures into separate figures for mucus and lumen? Also state p and R-values.
Table 1: How was the confounding factors determined? By Permanova? Consider doing e.g. a canonical correspondance analysis (rCCA) also/instead.
Suppl. Fig. 4 indicate a strong run-effect. Was this taken into account (e..g by randomising samples to different runs) when doing the analysis (DGGE)? Was PCR1 for amplicon-sequencing done in one run or is there also a potential run-effect/systematic effect influencing data interpretation here?

Additional comments

Interesting study addressing a very relevant research question.

Specific comments, not already covered above:
Line 19: Add "dysbiosis" after "gut microbiota"
Line 31: Consider "softening" the conclusion to "our findings strongly indicate that..."
Line 72: What is meant by lower abundance? Absolute abundance? Relative abundance of certain taxa?
Line 251-255: Are these findings significant? Could there be an run-effect as discussed above?
All over the manuscript: "Illumina sequencing" is a very vague and unspecific term. Be more specific and call it e.g. "high-throughput 16S rRNA gene amplicon sequencing"
Several places: Phyla-names are not in italics

---

## Round 0.2 · accepted · Accept

I agree with both reviewers that you did a very good job in incorporating all suggestions. Very nice paper indeed!

·

Basic reporting

I am happy with the changes made to the manuscript, particularly regarding the improvement of the language, and enhanced clarity regarding methods.

Experimental design

no comment

Validity of the findings

no comment

Reviewer 2 ·

Basic reporting

All OK in the revised version of the manuscript.

One small remark: In line 76 "microbiota" should be exchanged with e.g. "taxa".

Experimental design

All OK in the revised version of the manuscript.

Validity of the findings

All OK in the revised version of the manuscript.

Additional comments

The revised version is much improved and acceptable for publication in PeerJ